# TEMPORAL DIFFERENCE LEARNING FOR DIFFUSION MODELS

## ABSTRACT

Diffusion models are typically trained with reconstruction losses at single, isolated time steps, which does not enforce consistency between predictions along the denoising trajectory. This lack of cross-time consistency can degrade performance, especially for few-step samplers. We introduce a temporal difference (TD) objective that penalizes inconsistency of the model's *multi-step* progress along the denoising path. By reformulating the diffusion process as a Markov reward process and casting the denoising task as a policy evaluation problem in reinforcement learning, we derive a unified TD approach that applies to both discrete- and continuous-time diffusion formulations. We further propose a principled sample-based reweighting method that stabilizes training.

Empirically, we show that adding our TD objective can significantly improve sample efficiency and enhance generative model quality, as measured by FID. In particular, TD exhibits stronger advantages when the number of sampling steps is small, highlighting its practical utility under low-computation-budget scenarios. We provide extensive ablation studies to justify our design choices, including loss reweighting, regularization weight, and one-step distance. Overall, our TD approach can be a general drop-in that enforces cross-time consistency and improves fixed-NFE generation quality, with potential utility across a wide range of diffusion generative models.

## 1 INTRODUCTION

Diffusion models have become a standard tool for high-fidelity generative modeling across images, audio, and beyond (Ho et al., 2020; Song et al., 2021b; Karras et al., 2022). Despite impressive progress in *sampler* design (e.g., probability-flow ODE/DDIM (Song et al., 2021a), high-order solvers such as DPM-Solver (Lu et al., 2022) and UniPC (Zhao et al., 2023)) and *training-time accelerations* (e.g., progressive distillation (Salimans & Ho, 2022) and consistency-style learning (Song et al., 2023)), the dominant training paradigm still optimizes *single-time* reconstruction/noise-prediction losses.

Such single-time objectives do not explicitly require that predictions made at different noise levels form a *time-consistent* trajectory under the known forward corruption process. The resulting cross-time mismatch can accumulate along the denoising path and becomes particularly detrimental when the sampler uses few steps (small NFE), where local errors have limited opportunity to average out (Song et al., 2021b; Karras et al., 2022). This motivates treating diffusion training through the lens of sequential decision making, where predictions at different timesteps must be consistent over multiple steps rather than only locally accurate.

Recent years have seen a surge of works that formulate diffusion sampling as a multi-step decision problem and apply reinforcement learning (RL) to optimize non-differentiable rewards. DDPO casts denoising as a Markov decision process (MDP) and shows policy-gradient updates can align text-to-image models with black-box objectives such as aesthetics and compressibility (Black et al., 2024). DPOK performs online RL with KL regularization to fine-tune diffusion models from human-trained reward functions, improving both alignment and fidelity (Fan et al., 2023).

Subsequent work explores algorithmic variants: LOOP analyzes the efficiency–performance trade-off between REINFORCE and PPO and proposes a leave-one-out PPO scheme for diffusion fine-tuning (Gupta et al., 2025); SEPO develops a policy-gradient method for discrete diffusion with

theoretical justification and strong results across discrete generative tasks (Zekri et al., 2025). Beyond images, RL fine-tuning has been applied to diffusion TTS via a loss-guided policy optimization objective (DLPO) (Chen et al., 2024). Other directions include self-play (SPIN-Diffusion), where a model competes with its past checkpoints to iteratively improve under a reward signal (Yuan et al., 2024), and forward-process RL that integrates reinforcement signals into flow-/score-matching objectives for online fine-tuning (Zheng et al., 2025).

Finally, at the theoretical level, Temporal Difference (TD) Flows connect TD learning with flow-based training, providing an RL interpretation of generative flows (Farebrother et al., 2025). Earlier work on generative TD learning proposed the $\gamma$-model framework, which reinterprets TD updates as a generative modeling problem for infinite-horizon prediction (Janner et al., 2020). While these methods highlight the synergy between RL and generative models, they focus either on flow-based models or on predictive state distributions.

A complementary strand leverages diffusion *as the policy class* for decision making in RL: Diffuser denoises entire trajectories to plan behaviors (Janner et al., 2022), Diffusion-QL represents policies with conditional diffusion models for offline RL (Wang et al., 2023), Diffusion Policy learns visuomotor control via action diffusion (Chi et al., 2023), and hierarchical methods introduce subgoal-conditioned diffusion for long-horizon tasks (Li et al., 2023). These approaches focus on maximizing environmental returns in external tasks.

In contrast, our work performs *policy evaluation* over the denoising process itself. We reformulate diffusion as a Markov reward process (MRP) and introduce a TD objective that enforces multi-step cross-time consistency of predictions along the denoising path, unified across discrete- and continuous-time formulations. To stabilize optimization across different time pairs, we further propose a principled sample-based loss reweighting scheme that equalizes loss scales. Rather than steering outputs via task-specific rewards, our TD formulation serves as a general-purpose training that improves fixed-NFE generation by aligning the model's internal temporal dynamics.

In Sec. 2, we review background and notation, including a unified two-time-mean form. Sec. 3 presents our TD objective (discrete derivation, unified form, design rules, and case for EDM). For the experiment results we present in Sec. 4 and discuss our method in Sec. 5. The appendices include implementation details, scheduler definitions, and additional proofs.

## 2 BACKGROUND AND NOTATION

Diffusion models corrupt data with a forward (noising) process and learn a denoiser or score function that inverts it (Ho et al., 2020; Song et al., 2021b). Discrete-time formulations such as DDPM (Ho et al., 2020; Nichol & Dhariwal, 2021) and the deterministic DDIM (Song et al., 2021a) provide simple training objectives and flexible sampling schedules, while continuous-time formulations based on differential equations (ODE/SDE) unify these views and support principled SDE samplers (Song et al., 2021b). To address the complex design space of diffusion models, EDM (Karras et al., 2022) modularizes the design space, refines several design choices (e.g., noise grids, preconditioning, loss weighting, etc.) and achieves strong empirical results with few-step sampling. In this work, we adopt these foundations and focus on enforcing *cross-time consistency* at training time.

### 2.1 UNIFIED TWO-TIME POSTERIOR MEAN

Let $t$ denote the time index: a discrete time $t \in \{0, \dots, T\}$ (DDPM/DDIM), or a continuous time $t \in [0, T]$ (ODE/SDE). Let $\tau < t$ be an *earlier (cleaner)* time point. Across families, the (true) mean of the posterior $q(\boldsymbol{x}_\tau | \boldsymbol{x}_t, \boldsymbol{x}_0)$ admits the *same linear form* (derived in Appendix B)

$$\boldsymbol{\mu}_\tau^{\text{true}}(\boldsymbol{x}_t, \boldsymbol{x}_0) \;=\; A_{t,\tau}\, \boldsymbol{x}_0 \;+\; \kappa_{t,\tau}\, \boldsymbol{x}_t, \tag{1}$$

where $\boldsymbol{x}_0$ is a clean datum (e.g., original image) and $\boldsymbol{x}_t$ is a noisy sample at level $t$. Family-specific choices for $(A_{t,\tau}, \kappa_{t,\tau})$ are summarized in Table 1. This linear form will allow us to define a surrogate mean by replacing $\boldsymbol{x}_0$ with the model prediction $D_\theta(\boldsymbol{x}_t; t)$ where $D_\theta$ is a model parametrized by $\theta$. This will be the main strategy of our TD objective (Sec.3).

| Process | $A_{t,\tau}$ | $\kappa_{t,\tau}$ |
|---|---|---|
| **DDPM** (Ho et al., 2020) | $\dfrac{\sqrt{\bar{\alpha}_\tau}\,(1-\alpha_t)}{1-\bar{\alpha}_t}$ | $\dfrac{\sqrt{\alpha_t}\,(1-\bar{\alpha}_\tau)}{1-\bar{\alpha}_t}$ |
| **DDIM** (Song et al., 2021a) | $\sqrt{\bar{\alpha}_\tau} - \sqrt{\dfrac{1-\bar{\alpha}_\tau}{1-\bar{\alpha}_t}}\,\sqrt{\bar{\alpha}_t}$ | $\sqrt{\dfrac{1-\bar{\alpha}_\tau}{1-\bar{\alpha}_t}}$ |
| **VP-SDE** (Song et al., 2021b) | $\alpha(\tau) - \kappa_t\,\alpha(t)$ | $\dfrac{\alpha(t)}{\alpha(\tau)} \cdot \dfrac{1-\alpha(\tau)^2}{1-\alpha(t)^2}$ |
| **VE-SDE** (Song et al., 2021b), **EDM** (Karras et al., 2022) | $1 - \dfrac{\sigma(\tau)^2}{\sigma(t)^2}$ | $\dfrac{\sigma(\tau)^2}{\sigma(t)^2}$ |

Table 1: Unified mean $\boldsymbol{\mu}_\tau^{\text{true}}(\boldsymbol{x}_t, \boldsymbol{x}_0) = A_{t,\tau}\boldsymbol{x}_0 + \kappa_{t,\tau}\boldsymbol{x}_t$ across diffusion families. For DDPM/DDIM, $\tau$ should be $t-1$ (i.e., the previous time step). For VP/VE/EDM, $(t,\tau)$ are continuous time indices (or equivalently, noise levels) selected by the sampler. Detailed definition of parameter and derivation of formulation for each models see Appendix B.

**EDM-style preconditioning.** Following EDM (Karras et al., 2022), we wrap the raw network $F_\theta$ in the preconditioned denoiser[1]

$$D_\theta(\boldsymbol{x}; t) = c_{\text{skip}}(t)\,\boldsymbol{x} + c_{\text{out}}(t)\,F_\theta\big(c_{\text{in}}(t)\,\boldsymbol{x};\ c_{\text{noise}}(t)\big), \tag{2}$$

and (for EDM) train with a weighted regression

$$\mathcal{L}_{\text{EDM}} = \mathbb{E}_{t, \boldsymbol{x}_0, \boldsymbol{x}_t}\Big[w(t)\,\big\|D_\theta(\boldsymbol{x}_t; t) - \boldsymbol{x}_0\big\|_2^2\Big]. \tag{3}$$

The specific choices of $\{c_{\text{skip}}, c_{\text{out}}, c_{\text{in}}, c_{\text{noise}}, w(t)\}$ equalize the effective variance across noise levels and improve optimization conditioning (Karras et al., 2022). We will develop a weighting to achieve a similar effect later, and combine Equation (3) with our TD objective.

## 2.2 TEMPORAL DIFFERENCE LEARNING

In this work, we will recast training diffusion models as policy evaluation problems in reinforcement learning (RL). A finite-horizon Markov reward process (MRP) (Szepesvari, 2010) is characterized by a tuple $(\mathcal{X}, r_t, P_t, T)$ where $\mathcal{X}$ is the (common) state space, $r_t : \mathcal{X} \times \mathcal{X} \mapsto \mathbb{R}$ is the reward function at time $t$, $P_t : \mathcal{X} \mapsto \Delta(\mathcal{X})$ is the transition kernel at time $t$ and $T$ is the length of the episode. However, to match the time notation in diffusion models, here we let the MRP start from step $t = T$ and traverse backward to $t = 0$. Specifically, when transitioning from $\boldsymbol{x}_t$ to $\boldsymbol{x}_{t-1}$ according to $P_t(\cdot|\boldsymbol{x}_t)$, we receive a reward of $r_{t-1}(\boldsymbol{x}_t, \boldsymbol{x}_{t-1})$. The state value function is then defined as the expected return when starting from a given state:

$$v_t(\boldsymbol{x}) := \mathbb{E}\left[G_t \mid \boldsymbol{x}_t = \boldsymbol{x}\right] := \mathbb{E}\left[\sum_{i=1}^{t} r_{t-i} \,\middle|\, \boldsymbol{x}_t = \boldsymbol{x}\right] \tag{4}$$

where $r_{t-i}$ is short for $r_{t-i}(\boldsymbol{x}_{t-i+1}, \boldsymbol{x}_{t-i})$ and the expectation is taken over $P_i$ for $0 < i \leq t$.

One can use temporal difference (TD) learning (Sutton et al., 1998) to find $v_t$. Given some parametrized functions $v_{\theta,t}$, the TD error of the transition $(\boldsymbol{x}_t, \boldsymbol{x}_{t-1})$ is

$$\delta_t := r_{t-1} + v_{\theta,t-1}(\boldsymbol{x}_{t-1}) - v_{\theta,t}(\boldsymbol{x}_t). \tag{5}$$

By minimizing the TD error using samples from the MRP and semi-gradient descent, $v_{\theta,t}$ will converge to the true $v_t$ (Tsitsiklis & Van Roy, 1996).

## 3 TEMPORAL DIFFERENCE LEARNING FOR DIFFUSION MODELS

This section develops TD training for diffusion models. We first derive the TD loss on a discrete time index grid, then lift it to a unified formulation that applies to both discrete (DDPM/DDIM) and continuous-time (VP/VE/EDM) families. Whenever a posterior mean is needed, we *do not* restate family-specific formulas – rather, we directly use the unified form in Equation (1).

---

[1]Note that EDM adopts a noise scheme instead of a time scheme, but we use time here as they are equivalent and it can facilitate later discussion.

### 3.1 DISCRETE TIME: MRP FORMULATION FOR DDPM

**MRP specifications on the time index grid.** Following Sec.2.2, we define an MRP, traversing backward in time from $t = T$ to $t = 0$, as follows:

- *State space:* $\mathcal{X}$ is the data space. In the case of image generation, $\mathcal{X}$ is the set of images.
- *Reward function:* $r_{t-1} := r_{t-1}(x_t, x_{t-1}) := \mu_{t-1}^{\text{true}}(x_t, x_0) - \mu_{t-2}^{\text{true}}(x_{t-1}, x_0)$ is the posterior mean difference. Note that $r_{t-1}$ is a vector in the data space, as opposed to a scalar reward common in reinforcement learning. This corresponds to a multiple-reward setting, which also changes the corresponding notations later such as return and state value. In the final step when transitioning from $x_1$ to $x_0$, the reward is defined as $r_0 := \mu_0^{\text{true}}(x_1, x_0) - x_0 = 0$. The last equation is because $\mu_0^{\text{true}}(x_1, x_0)$, the conditional mean, must be $x_0$ *given* $x_0$.
- *Transition kernel:* $P_t(x_{t-1} \mid x_t) := q(x_{t-1}|x_t, x_0)$ is induced by the posterior of the predecessor in the diffusion process (stochastic in $x_{t-1}$ through the forward coupling).

**Return, value, TD(0) and $k$-step return.** With the MRP setup, the return $g_t$ is the displacement from data:

$$g_t = \mu_{t-1}^{\text{true}}(x_t, x_0) - x_0, \tag{6}$$

and it satisfies the normal recursion (although backward in time) $g_t = r_{t-1} + g_{t-1}$. Moreover, it ensures that $g_1 = \mu_0^{\text{true}}(x_1, x_0) - x_0 = x_0 - x_0 = 0$. For this return, there is no randomness given $x_t$ and $x_0$, which means the state value is the same as the return itself

$$v_t(x_t) := \mathbb{E}[g_t \mid x_t, x_0] = \mu_{t-1}^{\text{true}}(x_t, x_0) - x_0. \tag{7}$$

The $t$ in $v_t$ corresponds to the return $g_t$ in time step $t$, while the $t$ in $x_t$ (or the $x_t$ itself) is for the condition in the expectation. We approximate $v_t$ using the preconditioned denoiser $D_\theta$ in (2):

$$v_t(x_t) \approx v_{\theta,t}(x_t) := \mu_{\theta,t-1}(x_t) - x_0, \qquad \mu_{\theta,t-1}(x_t) := A_{t,t-1} D_\theta(x_t; t) + \kappa_{t,t-1} x_t. \tag{8}$$

To learn the parameters $\theta$, we construct a bootstrap target with fixed parameters. The "next"-state's value is estimated by

$$v_{t-1}(x_{t-1}) \approx v_{\theta',t-1}(x_{t-1}) = \mu_{\theta',t-2}(x_{t-1}) - x_0, \tag{9}$$

$$\mu_{\theta',t-2}(x_{t-1}) = A_{t-1,t-2} D_{\theta'}(x_{t-1}; t-1) + \kappa_{t-1,t-2} x_{t-1}. \tag{10}$$

where $\theta'$ is the fixed (stop-gradient) target network's parameters, updated using Polyak averaging. By the definition of the reward, the bootstrap target is

$$r_{t-1} + v_{\theta',t-1}(x_{t-1}) = \mu_{t-1}^{\text{true}}(x_t, x_0) - \mu_{t-2}^{\text{true}}(x_{t-1}, x_0) + \mu_{\theta',t-2}(x_{t-1}) - x_0 \tag{11}$$

Then the TD error is

$$\delta_t = r_{t-1} + v_{\theta',t-1}(x_{t-1}) - v_{\theta,t}(x_t) \tag{12}$$

$$= \mu_{t-1}^{\text{true}}(x_t, x_0) - \mu_{t-2}^{\text{true}}(x_{t-1}, x_0) + \mu_{\theta',t-2}(x_{t-1}) - \mu_{\theta,t-1}(x_t) \tag{13}$$

$$= \underbrace{\left[\mu_{t-1}^{\text{true}}(x_t, x_0) - \mu_{t-2}^{\text{true}}(x_{t-1}, x_0)\right]}_{\text{one-step diffusion drift}} - \underbrace{\left[\mu_{\theta,t-1}(x_t) - \mu_{\theta',t-2}(x_{t-1})\right]}_{\text{one-step model drift}}. \tag{14}$$

The TD(0) objective is then

$$\mathcal{L}_{\text{TD}(0)} = \mathbb{E}_{t \sim \mathcal{U}\{2,T\}, x_0 \sim q_{\text{data}}(x_0), x_{t-1} \sim q(x_{t-1}|x_0), x_t \sim q(x_t|x_{t-1})} \left[\|\delta_t\|_2^2\right]. \tag{15}$$

Minimizing the TD error can be interpreted as aligning diffusion progression across time steps as shown in Equation (14). If the true mean has drifted in one step, the model should shift in the same way, thus enforcing *consistency* between time steps. The same derivation applies to DDIM (Song et al., 2021a) by substituting its $(A_{t,t-1}, \kappa_{t,t-1})$ from Table 1.

We can also use $k$-step return as the bootstrap target. Keep expanding Equation (11) over $k$ steps gives the following $k$-step objective

$$\mathcal{L}_{\text{TD}}^{(k)} = \mathbb{E}\left[\|\underbrace{\left[\mu_{t-1}^{\text{true}}(x_t, x_0) - \mu_{t-k-1}^{\text{true}}(x_{t-k}, x_0)\right]}_{k\text{-step diffusion drift}} - \underbrace{\left[\mu_{\theta,t-1}(x_t) - \mu_{\theta',t-k-1}(x_{t-k})\right]}_{k\text{-step model drift}}\|_2^2\right] \tag{16}$$

where the expectation is over $t \sim \mathcal{U}\{k+1, T\}, x_0 \sim q_{\text{data}}(x_0), x_{t-k} \sim q(x_{t-k}|x_0), x_t \sim q(x_t|x_{t-k})$.

**Aligning model with posterior.** Although our TD objective can encourage temporal consistency w.r.t. the diffusion drift, it may be insufficient in aligning the model with the true posterior by itself. Thus, we combine it with the original DDPM loss

$$\mathcal{L}_{\mathrm{DDPM}} \; = \; \mathbb{E}\Big[w(t)\|\boldsymbol{\mu}_{t-1}^{\mathrm{true}}(\boldsymbol{x}_t, \boldsymbol{x}_0) - \boldsymbol{\mu}_{\theta,t-1}(\boldsymbol{x}_t)\|_2^2\Big] \tag{17}$$

with some weighting $w(t)$ and the final objective is

$$\mathcal{L}_{\mathrm{TD+DDPM}}^{(k)} \; = \; \mathcal{L}_{\mathrm{TD}}^{(k)} \; + \; \lambda\,\mathcal{L}_{\mathrm{DDPM}} \tag{18}$$

where $\lambda > 0$ is a hyperparameter. Finally, note that there is no need to bootstrap when $t \leq k$ because we do know the rest of the episode and the corresponding true return. In this case, the TD loss actually coincides with the DDPM loss, and the objective $\mathcal{L}_{\mathrm{TD+DDPM}}^{(k)}$ is equivalent to $(1+\lambda)\mathcal{L}_{\mathrm{DDPM}}$. In other words, if we sampled a time $t > k$ during training, the TD loss can be used and we optimize $\mathcal{L}_{\mathrm{TD+DDPM}}^{(k)}$, otherwise, we optimize $(1+\lambda)\mathcal{L}_{\mathrm{DDPM}}$.

### 3.2 Discrete and Continuous Time: A Unified TD Objective

The derivations based on discrete time steps above can be easily extended to continuous-time scenarios with ODE/SDE thanks to the unified mean (Equation (1) from Sec. 2.1).

Instead of matching drifts that are $k$-step away in the discrete case, here we pick two time indices $t, t' \in [0, T]$. Time $t$ (resp. $t'$) induces a true posterior mean for an earlier time $\tau < t$ (resp. $\tau' < t'$). Their corresponding means can be expressed as

$$\boldsymbol{\mu}_{\tau}^{\mathrm{true}}(\boldsymbol{x}_t, \boldsymbol{x}_0) \; = \; A_{t,\tau}\,\boldsymbol{x}_0 \; + \; \kappa_{t,\tau}\,\boldsymbol{x}_t \qquad \boldsymbol{\mu}_{\tau'}^{\mathrm{true}}(\boldsymbol{x}_{t'}, \boldsymbol{x}_0) \; = \; A_{t',\tau'}\,\boldsymbol{x}_0 \; + \; \kappa_{t',\tau'}\,\boldsymbol{x}_{t'}. \tag{19}$$

For the discrete-time case (Equation (16)), $\boldsymbol{\mu}_{t-1}^{\mathrm{true}}(\boldsymbol{x}_t, \boldsymbol{x}_0)$ is the posterior mean in the previous time step (i.e., the subscript $t-1$). In analogy, $\tau$ here can be considered as the "previous time step" of $t$ in the continuous case. In our experiments, we set $0 \leq \tau' < t' < \tau < t \leq T$ with *span* $k := t - t'$ and *stride* $\Delta := t' - \tau' = t - \tau < k$, imitating the discrete case. Accordingly, the model is defined as

$$\boldsymbol{\mu}_{\theta,\tau}(\boldsymbol{x}_t) \; := \; A_{t,\tau}\,D_\theta(\boldsymbol{x}_t; t) \; + \; \kappa_{t,\tau}\,\boldsymbol{x}_t, \tag{20}$$

and the TD loss for continuous time reads

$$\mathcal{L}_{\mathrm{TD}}^{\mathrm{cont}} = \mathbb{E}_{\boldsymbol{x}_0, t, t', \boldsymbol{x}_t, \boldsymbol{x}_{t'}}\Big[\big\|\big[\boldsymbol{\mu}_{\tau}^{\mathrm{true}}(\boldsymbol{x}_t, \boldsymbol{x}_0) - \boldsymbol{\mu}_{\tau'}^{\mathrm{true}}(\boldsymbol{x}_{t'}, \boldsymbol{x}_0)\big] - \big[\boldsymbol{\mu}_{\theta,\tau}(\boldsymbol{x}_t) - \boldsymbol{\mu}_{\theta',\tau'}(\boldsymbol{x}_{t'})\big]\big\|_2^2\Big] \tag{21}$$

In the following, we will use EDM (Karras et al., 2022) as an example and show how our TD objective can be integrated into training continuous-time diffusion models.

**Pairwise loss weighting.** When using the preconditioned denoiser $D_\theta$ in the form of (2), we can adjust the loss of any sampled pair $(t, t')$ so that the loss scales of different pairs are not affected by the choices of span $k$, stride $\Delta$ and output scaling $c_{\mathrm{out}}$. First note that the TD error in (21) can be expressed as

$$\boldsymbol{\delta}_{t,t'} := \big[\boldsymbol{\mu}_{\tau}^{\mathrm{true}}(\boldsymbol{x}_t, \boldsymbol{x}_0) - \boldsymbol{\mu}_{\tau'}^{\mathrm{true}}(\boldsymbol{x}_{t'}, \boldsymbol{x}_0)\big] - \big[\boldsymbol{\mu}_{\theta,\tau}(\boldsymbol{x}_t) - \boldsymbol{\mu}_{\theta',\tau'}(\boldsymbol{x}_{t'})\big] \tag{22}$$

$$= \big[A_{t,\tau}\boldsymbol{x}_0 - A_{t',\tau'}\boldsymbol{x}_0\big] - \big[A_{t,\tau}D_\theta(\boldsymbol{x}_t, t) - A_{t',\tau'}D_{\theta'}(\boldsymbol{x}_{t'}, t')\big] \tag{23}$$

which is due to Equations (19) and (20). Using Equation (2), and denoting $F_{\theta,t}(\boldsymbol{x}_t) := F_\theta\big(c_{\mathrm{in}}(t)\,\boldsymbol{x}_t;\; c_{\mathrm{noise}}(t)\big)$, $F_{\theta',t'}(\boldsymbol{x}_{t'}) := F_{\theta'}\big(c_{\mathrm{in}}(t')\,\boldsymbol{x}_{t'};\; c_{\mathrm{noise}}(t')\big)$, the TD error can be further decomposed into

$$\boldsymbol{\delta}_{t,t'} = \big[A_{t,\tau}\boldsymbol{x}_0 - A_{t',\tau'}\boldsymbol{x}_0\big] - \big[A_{t,\tau}c_{\mathrm{skip}}(t)\,\boldsymbol{x}_t - A_{t',\tau'}c_{\mathrm{skip}}(t')\,\boldsymbol{x}_{t'}\big] \tag{24}$$

$$- \big[A_{t,\tau}c_{\mathrm{out}}(t)F_{\theta,t}(\boldsymbol{x}_t) - A_{t',\tau'}c_{\mathrm{out}}(t')F_{\theta',t'}(\boldsymbol{x}_{t'})\big] \tag{25}$$

Since $F_\theta, F_{\theta'}$ are the raw models, minimizing the TD error is effectively minimizing the following normalized errors

$$\boldsymbol{e}_t := \frac{\boldsymbol{x}_0 - c_{\mathrm{skip}}(t)\boldsymbol{x}_t}{c_{\mathrm{out}}(t)} - F_{\theta,t}(\boldsymbol{x}_t) \qquad \boldsymbol{e}_{t'} := \frac{\boldsymbol{x}_0 - c_{\mathrm{skip}}(t')\boldsymbol{x}_{t'}}{c_{\mathrm{out}}(t')} - F_{\theta',t'}(\boldsymbol{x}_{t'}), \tag{26}$$

rescaled by $A_{t,\tau}c_{\text{out}}(t)$ (resp. $A_{t',\tau'}c_{\text{out}}(t')$). That is, $\boldsymbol{\delta}_{t,t} = \mathcal{B}\,\boldsymbol{e}_{t,t'}$ where

$$\mathcal{B} := [A_{t,\tau}c_{\text{out}}(t)\cdot I,\ -A_{t',\tau'}c_{\text{out}}(t')\cdot I] \in \mathbb{R}^{d\times 2d} \quad \text{and} \quad \boldsymbol{e}_{t,t'} := \begin{bmatrix} \boldsymbol{e}_t \\ \boldsymbol{e}_{t'} \end{bmatrix} \in \mathbb{R}^{2d}. \quad (27)$$

Then we can see that

$$\|\boldsymbol{\delta}_{t,t'}\|_2^2 \leq \|\mathcal{B}\|_2^2\cdot\|\boldsymbol{e}_{t,t'}\|_2^2 = (A_{t,\tau}^2 c_{\text{out}}(t)^2 + A_{t',\tau'}^2 c_{\text{out}}(t')^2)\cdot\|\boldsymbol{e}_{t,t'}\|_2^2 \quad (28)$$

Since $\|\boldsymbol{e}_{t,t'}\|_2^2$ is the normalized error w.r.t. the raw models $F_\theta, F_{\theta'}$, it would be helpful to set a uniform scale that is not affected by the choices of span $k$, stride $\Delta$ and output scaling $c_{\text{out}}$, which leads to the following pairwise weighting:

$$w_{\text{TD}}(t,t') = \frac{1}{A_{t,\tau}^2 c_{\text{out}}(t)^2 + A_{t',\tau'}^2 c_{\text{out}}(t')^2}. \quad (29)$$

Then $w_{\text{TD}}(t,t')\|\boldsymbol{\delta}_{t,t'}\|_2^2 \leq \|\boldsymbol{e}_{t,t'}\|_2^2$, and the weighted TD objective is

$$\mathcal{L}_{\text{wTD}}^{\text{cont}} = \mathbb{E}_{\boldsymbol{x}_0,t,t'}\Big[w_{\text{TD}}(t,t')\big\|\boldsymbol{\delta}_{t,t'}\big\|_2^2\Big]. \quad (30)$$

**Aligning model with posterior.** Similar to the discrete case, $\mathcal{L}_{\text{wTD}}^{\text{cont}}$ alone may be insufficient to align the model with the posterior. Therefore, we combine it with the original EDM reconstruction loss (3) and optimize

$$\mathcal{L}_{\text{TD+EDM}} = \mathcal{L}_{\text{wTD}}^{\text{cont}} + \lambda\mathcal{L}_{\text{EDM}}. \quad (31)$$

**Sampling scheme for time index.** Instead of a time schedule for $t$, EDM adopts a noise schedule for $\sigma$ that can potentially produce a very large variance. In fact, during training, the noise level is sampled from a log-normal distribution, which means the noise is unbounded during training. However, during data sampling, EDM applies Heun's 2nd order solver on a noise grid specified by

$$\sigma(i) = \left(\sigma_{\max}^{1/\rho} + \frac{i}{N-1}\big(\sigma_{\min}^{1/\rho} - \sigma_{\max}^{1/\rho}\big)\right)^\rho, \quad i \in \{0,\dots,N-1\} \quad (32)$$

where $N = 18$ is the grid size, $\rho = 7$, $\sigma_{\max} = 80$ and $\sigma_{\min} = 0.002$ are the upper and lower bounds of the sampler. Since $[\sigma_{\min}, \sigma_{\max}]$ is the operating range of the sampler, the model should allocate/dedicate more capacity to predict well in this region. Therefore, for better temporal consistency, we include our TD loss when a $\sigma$ is sampled in this range during model training.

Specifically, during training, any $\sigma > 0$ can be sampled since it is log-normal. If the sampled $\sigma$ is in $[\sigma_{\min}, \sigma_{\max}]$, we can identify the corresponding $i$ (not necessarily an integer) from Equation (32). We treat this $i$ as the time index $t$ and find a different time step[2] $t' = t + k$ for the bootstrap, which corresponds to a different noise level $\sigma(t')$. Moreover, since the ODE solver operates on an integer grid (32), we set $k = 1$, encouraging "one-step" consistency. To find the posterior means, we set $\Delta$ to be a fraction of $k$ (e.g., $\Delta = k/3$) and calculate the means at $\tau = t + \Delta$ and $\tau' = t' + \Delta$. Finally, to make sure that even the noise level corresponding to $\tau'$ is in $[\sigma_{\min}, \sigma_{\max}]$, we apply the TD loss only when $\sigma(\tau')$ is also in $[\sigma_{\min}, \sigma_{\max}]$ and optimize (31), otherwise, we optimize $(1 + \lambda)\mathcal{L}_{\text{EDM}}$ similar to the discrete-time case. The algorithm is summarized in Algorithm 1.

## 4 EMPIRICAL STUDY

This section aims to answer the following questions: 1. Performance and Applicability: Can our TD approach outperform the traditional EDM on the CIFAR-10 benchmark using a standard-sized network and limited hyperparameter tuning? Under what conditions does our method demonstrate particular advantages? 2. Ablation Study: Are the design choices in our TD algorithm effective? Specifically: 2.1) In our combined TD+EDM objective, does a larger weight $\lambda$ on the EDM loss consistently yield better results? 2.2) How sensitive is the model's performance to the fractional stride hyperparameter $\Delta$? 2.3) How effective is our proposed sample-wise weighting mechanism?

We evaluate on **CIFAR-10** ($32\times32$) (Krizhevsky et al., 2009) using FID-50k against train statistics. Samplers use the probability-flow ODE with the Heun integrator; *NFE* $= 2 \times$ steps $- 1$. Unless specified otherwise, we use the following *default TD setting*: $\Delta = 0.25, k = 1, \lambda = 0.5$ and TD weighting $w_{\text{TD}}$ in Equation (29). We refer readers to App. A for any missing details.

---

[2]We use $t' = t + k$ instead of $t' = t - k$ because then $t'$ will correspond to a smaller noise level, thus closer to the real data as common in the ODE/SDE formulations.

---

**Algorithm 1** TD + EDM training (per mini-batch)

---

**Input**: EDM denoiser $D_\theta$ in Equation (2); target network $\theta'$; EDM grid $\sigma(i)$; span $k$; stride $\Delta$; mixing coefficient $\lambda$.

1: Sample $\boldsymbol{x}_0 \sim q_{\text{data}}(\boldsymbol{x}_0)$ and $\ln \sigma \sim \mathcal{N}(P_{\text{mean}}, P_{\text{std}}^2)$.
2: $L_{\text{EDM}} = w(\sigma)\|D_\theta(\boldsymbol{x}_\sigma; \sigma) - \boldsymbol{x}_0\|_2^2$
3: **if** $\sigma \in [\sigma_{\min}, \sigma_{\max}]$ **then**
4:     Compute the corresponding $i$ from (32) and set $t = i$.
5:     **if** $t \leq N - 1 - k - \Delta$ (equivalent to $\sigma(\tau') \geq \sigma_{\min}$) **then**
6:         $L_{\text{TD}} = w_{\text{TD}}(t, t')\|\left[\boldsymbol{\mu}_\tau^{\text{true}}(\boldsymbol{x}_t, \boldsymbol{x}_0) - \boldsymbol{\mu}_{\tau'}^{\text{true}}(\boldsymbol{x}_{t'}, \boldsymbol{x}_0)\right] - \left[\boldsymbol{\mu}_{\theta,\tau}(\boldsymbol{x}_t) - \boldsymbol{\mu}_{\theta',\tau'}(\boldsymbol{x}_{t'})\right]\|_2^2$
7:         $L = L_{\text{TD}} + \lambda L_{\text{EDM}}$
8:     **else**
9:         $L = (1 + \lambda)L_{\text{EDM}}$
10:     **end if**
11: **else**
12:     $L = (1 + \lambda)L_{\text{EDM}}$
13: **end if**
14: Perform gradient descent on $L$ and update $\theta'$ using moving average.

---

### 4.1 PERFORMANCE AND APPLICABILITY

We conduct two experimental sets to answer the first question. The first set attempts to stay close to the standard EDM setting (Karras et al., 2022), testing whether our TD+EDM method surpasses vanilla EDM. The second series employs a smaller network to facilitate an extensive hyperparameter sweep. This investigation demonstrates that our approach holds a particular advantage when sampling is constrained by a limited computational budget.

In the first experiment, we adopt the default EDM configuration from (Karras et al., 2022), using a SongUNet trained with EDM preconditioning. We integrate our TD method using Equation (31). Each method is evaluated using a single training run with 18 sampling steps (i.e., *NFE* = 35). Figure 1 presents a qualitative comparison of samples generated by the EDM and our TD+EDM.

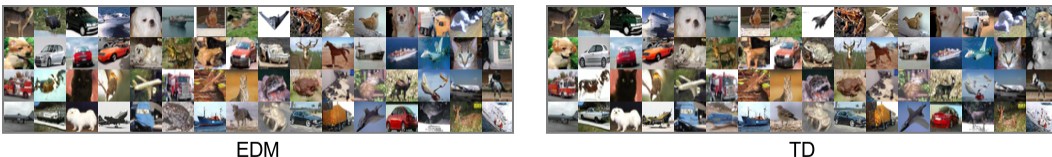

              EDM                                          TD

Figure 1: samples generated on CIFAR-10 under setting in Appendix.A. Both the baseline EDM model (left) and our TD+EDM model (right) were sampled using 18 steps (*NFE* = 35).

As shown in Table 2, TD+EDM yields a modest but consistent improvement over the vanilla EDM baseline. Specifically, the last-15% FID (i.e., average FID of the last 15% of the training) improves from 2.3875 to 2.3814 by using our approach, while the last-5% FID improves from 2.3893 to 2.3768. Although the absolute gains are small at this model capacity and NFE, the consistent improvement across both evaluation windows indicates that enforcing cross-time agreement does not impair performance and can slightly enhance the stability of end-of-training results. In contrast, EDM tends to show stable performance or even a mild degradation as training proceeds.

Table 2: CIFAR-10, large UNet, **steps=18** (*NFE* = 35). Default TD setting. The last 15% refers to the average FID score calculated over the final 15% of evaluations – specifically, the last 12 evaluations out of a total of 80.

| Method | last 15% average FID-50k ↓ | last 5% average FID-50k ↓ |
|---|---|---|
| EDM (baseline) | 2.3875 | 2.3893 |
| TD+EDM (ours) | 2.3814 | 2.3768 |

In the second experiment, we investigate the conditions under which our TD+EDM approach yields a clear advantage over EDM alone. As shown in Table 3, the benefit of our method becomes more pronounced as the number of diffusion steps (and thus the computational budget) decreases during sampling/generative stage. Note that this advantage remains true as we sweep over other hyperparamters as shown in the next section.

This advantage arises maybe because the TD objective shifts the optimization target from single-step denoising to multi-step trajectory consistency. Whereas conventional objectives act as local experts for ideal, small noise transitions (Song et al., 2023; Lu et al., 2022), our approach explicitly penalizes inconsistencies between direct and multi-step predictions. This training paradigm enhances robustness to the large discretization errors inherent in coarse sampling schedules (Karras et al., 2022), effectively training the model to anticipate and correct for future errors.

Table 3: The effects of weighting and sampling steps. Each number reports mean $\pm$ standard deviation over 3 seeds).

| Variant | steps=12 | steps=15 | steps=18 |
|---|---|---|---|
| Unweighted TD | $11.0589 \pm 0.1103$ | $10.5889 \pm 0.1547$ | $10.4352 \pm 0.1830$ |
| Weighted TD (Equation (29)) | $\mathbf{10.2244 \pm 0.1936}$ | $\mathbf{9.7547 \pm 0.1066}$ | $\mathbf{9.7512 \pm 0.0928}$ |
| EDM | $10.5755 \pm 0.1091$ | $10.2010 \pm 0.0555$ | $9.9776 \pm 0.0422$ |

## 4.2 ABLATION STUDY

Given compute constraints, all ablations use CIFAR-10 with a smaller UNet and report FID-50k averaged over three seeds. Unless a factor is being swept, we *anchor* hyperparameters at our final recipe $\Delta$=0.25, $k$=1, $\lambda$=0.5, with the *weighted* TD in Equation (29).

To answer question 2.1, we perform a sweep over the mixing coefficient $\lambda$. Note that since $\lambda$ weights the EDM loss component, a very high value would indicate a preference for the original EDM objective over the TD loss. As shown in Table 4, optimal performance across different sampling steps is consistently achieved with relatively small values of $\lambda$. Furthermore, performance remains stable across a range of these small $\lambda$ values for a given number of sampling steps. This result confirms the utility of the TD loss, as the EDM component must be downweighted for the best performance, and also indicates that the hyperparameter is convenient to tune, as performance is robust within a low range.

Table 4: Ablation on $\lambda$ (weighted). We vary $\lambda \in \{0.01, 0.5, 1.0, 2.0\}$ under the default $\Delta$=0.25, $k$=1, *weighted*. CIFAR-10 FID-50k $\downarrow$ (mean $\pm$ std over 3 seeds).

| $\lambda$ | steps=12 ($NFE = 23$) | steps=15 ($NFE = 29$) | steps=18 ($NFE = 35$) |
|---|---|---|---|
| 0.01 | $\mathbf{10.2118 \pm 0.0672}$ | $9.7931 \pm 0.1533$ | $\mathbf{9.7502 \pm 0.2190}$ |
| 0.50 | $10.2244 \pm 0.1936$ | $\mathbf{9.7547 \pm 0.1006}$ | $9.7512 \pm 0.0928$ |
| 1.00 | $10.3667 \pm 0.1165$ | $9.9768 \pm 0.1414$ | $9.6347 \pm 0.1700$ |
| 2.00 | $10.4133 \pm 0.1438$ | $9.9938 \pm 0.0783$ | $9.8383 \pm 0.1028$ |
| EDM | $10.5755 \pm 0.1091$ | $10.2010 \pm 0.0555$ | $9.9776 \pm 0.0422$ |

To address question 2.2, we investigate the sensitivity to the "one-step" length (stride) $\Delta$, a parameter introduced by our TD formulation. A key practical concern with TD methods is the introduction of new hyperparameters; we demonstrate utility by showing relative insensitivity to $\Delta$. As shown in Table 5, under a fixed number of sampling steps, varying $\Delta$ typically results in performance variations that are statistically insignificant. This indicates that the method is robust to the exact value of this parameter.

Table 5: Ablation on one-step $\Delta \in \{1/2, 1/3, 1/4, 1/5\}$ (weighted). CIFAR-10 FID-50k $\downarrow$ (mean $\pm$ std over 3 seeds).

| $\Delta$ | steps=12 ($NFE = 23$) | steps=15 ($NFE = 29$) | steps=18 ($NFE = 35$) |
|---|---|---|---|
| 1/2 | $10.3313 \pm 0.1829$ | $9.6981 \pm 0.1103$ | $\mathbf{9.6882 \pm 0.0818}$ |
| 1/3 | $10.2634 \pm 0.1060$ | $\mathbf{9.6343 \pm 0.1100}$ | $9.8003 \pm 0.0690$ |
| 1/4 | $\mathbf{10.2244 \pm 0.1936}$ | $9.7547 \pm 0.1066$ | $9.7512 \pm 0.0928$ |
| 1/5 | $10.2322 \pm 0.0556$ | $9.7870 \pm 0.1263$ | $9.7059 \pm 0.0540$ |
| EDM | $10.5755 \pm 0.1091$ | $10.2010 \pm 0.0555$ | $9.9776 \pm 0.0442$ |

To address question 2.3, we compare $w_{\mathrm{TD}}$ in Equation (29) against a constant weight 1, with $\Delta=0.25$, $k=1$, $\lambda=0.5$. This comparison is critical because the vanilla EDM employs a specific time-dependent weighting scheme to balance the loss across different noise levels, effectively biasing the training toward less frequently sampled time regions. Our TD loss introduces a distinct, principled weighting mechanism defined in Equation (29). As shown in Table 3, our weighting scheme provides a statistically significant advantage over the constant baseline when using different sampling steps.

## 5 DISCUSSIONS

We have introduced a TD learning framework for diffusion models that reformulates denoising across the time axis as a policy evaluation problem building upon the Markov reward process. Our primary contribution is a novel TD objective that explicitly penalizes inconsistencies in the model's predictions over multiple steps, thereby enforcing cross-time agreement. A key practical element is our derivation of a principled, sample-based loss reweighting scheme, $w_{\mathrm{TD}}$, which stabilizes training by bounding the loss scale across heterogeneous time pairs. Empirically, we establish the practical utility of our approach, showing that it provides a clear advantage in low-compute sampling regimes (12–18 steps) by reducing the accumulation of discretization errors. We further validate the robustness of our method through ablations on key hyperparameters, culminating in a practical recipe that consistently improves sample quality.

**Limitations**. While effective, our approach has several limitations. First, the TD objective introduces additional constant-factor computational and memory overhead, due to the update of the TD target neural network. Second, the empirical scope of this work is currently limited to unconditional image generation on CIFAR-10 at $32 \times 32$ resolution using a probability-flow ODE solver. While this scope is consistent with prior algorithmic studies in diffusion modeling (Ho et al., 2020; Karras et al., 2022), broader validation will be required for conditional and high-resolution settings. Finally, although we identify a robust mixing coefficient, the optimal balance between TD and reconstruction losses may depend on dataset, architecture, or even training stage.

**Future work**. These limitations motivate several promising research directions. The computational cost could be mitigated by amortizing target-network updates, employing lower-precision models, or using lightweight adapter modules. Another immediate next step is to extend validation to high-resolution conditional generation and to alternative sampler families (e.g., VP/VE). The framework also invites adaptive training curricula, such as annealing the mixing coefficient $\lambda$ or progressively widening the step span $k$ during training (Karras et al., 2022). Related approaches, such as SDPO, demonstrate the value of dense stepwise rewards for improving few-step sampling alignment (Zhang et al., 2024), and noise-correlation methods have also been explored to enhance temporal consistency in sequential diffusion settings (Lu et al., 2024). Finally, given its formulation as a drop-in objective, we anticipate strong synergies between our TD objective and other advances in diffusion, such as distillation for ultra-low NFE sampling (Salimans & Ho, 2022) and solver-aware training policies (Lu et al., 2022; Zhao et al., 2023), making it a versatile tool for improving temporal consistency in generative modeling.

ETHIC STATEMENT

This paper presents a methodological improvement for training diffusion models by enforcing temporal consistency. We acknowledge the significant ethical considerations associated with advancing generative modeling technology. Our work, which increases the sampling efficiency and quality of diffusion models, shares the dual-use potential inherent to this field: while it can empower beneficial applications in creative arts, design, and data augmentation, it could also lower the computational barrier for generating synthetic media that might be used to create misinformation or deepfakes.

We emphasize that our research is a theoretical contribution aimed at improving the understanding and performance of generative models. We strongly advocate for the development and use of such technologies to be accompanied by robust safeguards, including provenance standards, detection mechanisms, and public education, to mitigate potential harms. We believe the research community must engage proactively with these ethical challenges.

REPRODUCIBILITY STATEMENT

We try to ensure the reproducibility of our work. The main paper provides the algorithmic details, including pseudocode for direct implementation, clearly defined mathematical formulae for key quantities, and essential implementation details. The appendix contains comprehensive information necessary to replicate our experiments. All datasets used are publicly available and well-known. Upon publication, we will release a complete code repository with scripts to reproduce all experimental results.

LLM USAGE

We utilized Large Language Models (LLMs) as an assistive tool in preparing this manuscript. Their role was limited to editorial assistance, such as improving grammar and clarity, as well as technical help with LaTeX table formatting and supplementing part of our literature search. All core contributions, including the research ideas, methodological design, experiments, analysis, and interpretation of the findings, are the sole work of the authors.

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

# A EXTENDED EXPERIMENTAL DETAILS

This section provides full, implementation settings for all experiments. We separate the large-model comparison (**Part I**) and the small-model ablations (**Part II**).

## A.1 PART I: FULL SETTINGS (LARGE MODEL AT 18 STEPS)

**Datasets & preprocessing (EDM default setting).** CIFAR-10; official train/test split; resolution $32 \times 32$ for train/eval; normalized to $[-1, 1]$; AugmentPipe with $p=0.12$ (xflip, yflip, scale, rotate, aniso, translate); unconditional; total images seen: 200M ($\sim$4000 ticks).

**Model architecture (EDM default setting).** SongUNet (DDPM++), $\sim$62M parameters; base channels $ch=128$, multipliers $[2, 2, 2]$; residual blocks per level: 4; attention at resolution 16 with 1 head, head_dim 128; GroupNorm + SiLU; dropout $0.13$; time/noise embedding dims 512 (positional encoding); unconditional; output parameterization $\hat{x}_0$ (EDM preconditioning).

**EDM preconditioning & sampling (EDM default setting).** $\sigma_{\text{data}}=0.5$; EDM default preconditioning; log-noise sampling $P_{\text{mean}}= -1.2$, $P_{\text{std}}=1.2$; $\rho$-power grid $(\sigma_{\min}, \sigma_{\max}, \rho)=(2 \times 10^{-3}, 80, 7)$; training loss weight $w(\sigma)$: EDM default.

**TD objective settings.** "One-step" stride $\Delta=0.25$ (index unit); "$k$-step" span $k=1.0$; mixing $\lambda=0.5$; pairwise weighting $w_{\text{TD}}$ in Equation (29): on; coupling: Markov increments; EMA target network decay $\tau=0.999$, updated every step; boundary mask near $u \to 0$ with valid $i \in [0, N-1-k-\Delta]$; pair sampling: uniform on grid indices.

**Optimization and schedule (EDM default setting).** Adam, learning rate $2 \times 10^{-4}$, betas $(0.9, 0.999)$; no weight decay, no grad clip; constant LR, no warmup; batch 128 per GPU, global 512, no accumulation; precision fp32 (static scale 1); training length 200M images ($\sim$4000 ticks).

**Sampler & inference (EDM default setting).** Probability-flow ODE; Heun (order 2); steps 18 (NFE $= 2 \times 18-1 = 35$); $\rho$-power grid $(2 \times 10^{-3}, 80, 7)$; unconditional.

**Evaluation protocol.** FID-50k, deterministic evaluation with fixed seeds 0–49999.

**Compute & software.** $4 \times$H100 80GB

Table 6: Part I: key schedule and preconditioning constants.

| Item | Value |
|---|---|
| $\sigma_{\min}$, $\sigma_{\max}$, $\rho$ (sampling grid) | $2 \times 10^{-3}$, 80, 7 |
| $\sigma_{\text{data}}$ (preconditioning) | 0.5 |
| $P_{\text{mean}}$, $P_{\text{std}}$ (log-normal noise sampling) | $-1.2$, 1.2 |
| Loss weight $w(\sigma)$ | EDM default |

Table 7: Part I: TD regularizer settings.

| Item | Value |
|---|---|
| "One-step" stride (index unit) | 0.25 |
| "$k$-step" span $k$ | 1.0 |
| Mixing $\lambda$ in Eq. equation (31) | 0.5 |
| Pairwise weight $w_{\text{TD}}$ (Eq. equation (29)) | on |
| EMA teacher decay & freq. | 0.999; every step |
| Boundary masking window | valid $i \in [0, N-1-k-\Delta]$ |

## A.2 PART II: FULL SETTINGS (ABLATION EXPERIMENT)

**Datasets & preprocessing(EDM default setting).** CIFAR-10; official train/test split; resolution $32 \times 32$ for train/eval; normalized to $[-1, 1]$; AugmentPipe with $p=0.12$ (xflip, yflip, scale, rotate, aniso, translate); unconditional; total images seen: 120M ($\sim$2400 ticks).

**Model architecture.** SongUNet (DDPM++), $\sim$13M parameters; $ch=64$, multipliers $[1, 2, 2]$; 2 resblocks/level; attention at 16 with 1 head, head_dim 64; GroupNorm + SiLU; dropout 0.10; time/noise embed dims 256 (positional); unconditional; outputs $\hat{\boldsymbol{x}}_0$ (EDM).

**EDM preconditioning & sampling(EDM default setting).** $\sigma_{\text{data}}=0.5$; EDM default preconditioning; log-noise sampling $P_{\text{mean}}=-1.2$, $P_{\text{std}}=1.2$; $\rho$-power grid $(\sigma_{\text{min}}, \sigma_{\text{max}}, \rho)=(2\times10^{-3}, 80, 7)$; training loss weight $w(\sigma)$: EDM default.

**TD objective settings.** One-step $\Delta=0.25$ (index units); K-step $k=1.0$; mixing $\lambda=0.5$; pairwise weighting $w_{\text{TD}}$ in Eq. equation (29): on; coupling: Markov increments; EMA target network decay $\tau=0.999$, updated every step; boundary mask near $u\to0$ with valid $i \in [0, N-1-k-\Delta]$; pair sampling: uniform on grid indices.

**Optimization and schedule(EDM default setting).** Adam, LR $2\times10^{-4}$, betas $(0.9, 0.999)$; no weight decay, no grad clip; constant LR, no warmup; batch 256 per GPU (single-GPU runs), no accumulation; fp32 (static AMP); 120M images per experiment ($\sim$2400 ticks); seeds 0, 1, 2, 0; results averaged over 3 seeds/configurations.

**Sampler & inference(EDM default setting).** Probability-flow ODE; Heun (order 2); steps 12, 15 (NFE = 23, 29); $\rho$-power grid $(2\times10^{-3}, 80, 7)$; unconditional.

**Evaluation protocol.** FID-50k, deterministic evaluation with fixed seeds 0–49999.

**Compute & software.** NVIDIA RTX4090.

Table 8: Part II: key schedule and preconditioning constants.

| Item | Value |
| --- | --- |
| $\sigma_{\text{min}}$, $\sigma_{\text{max}}$, $\rho$ (sampling grid) | $2\times10^{-3}$, 80, 7 |
| $\sigma_{\text{data}}$ (preconditioning) | 0.5 |
| $P_{\text{mean}}$, $P_{\text{std}}$ (log-normal noise sampling) | $-1.2$, 1.2 |
| Loss weight $w(\sigma)$ | EDM default |

Table 9: Part II: TD regularizer settings.

| Item | Value |
| --- | --- |
| "One-step" stride (index unit) | $1/2, 1/3, 1/4, 1/5$ |
| "$k$-step" span $k$ | 1.0 |
| Mixing $\lambda$ in Eq. equation (31) | 0.01, 0.5, 1.0, 2.0 |
| Pairwise weight $w_{\text{TD}}$ (Eq. equation (29)) | weighted,unweighted |
| EMA teacher decay & freq. | 0.999; every step |
| Boundary masking window | valid $i \in [0, N-1-k-\Delta]$ |

## B FORWARD PROCESSES AND TWO-TIME MEANS

This appendix consolidates the forward/noising processes considered in the paper and rewrites their two-time posterior means in the unified linear form

$$\boldsymbol{\mu}_\tau^{\text{true}}(\boldsymbol{x}_t, \boldsymbol{x}_0) = A_{t,\tau}\,\boldsymbol{x}_0 + \kappa_{t,\tau}\,\boldsymbol{x}_t, \qquad \tau < t, \tag{33}$$

where $\boldsymbol{x}_0 \in \mathbb{R}^d$ is the clean datum and $\boldsymbol{x}_t$ is a noisy observation at level $t$ on the *same* time/noise axis as $\tau$. For discrete-time models (DDPM/DDIM), $t, \tau \in \{0, \ldots, T\}$ and we typically take $\tau = t-1$ on the native grid. For continuous-time models (VP/VE/EDM), $t, \tau \in [0, T]$ (or a monotone reparameterization such as $\sigma$) chosen by the sampler.

**Notation.** For DDPM/DDIM, set $\alpha_t = 1 - \beta_t$ and $\bar{\alpha}_t = \prod_{s=1}^{t} \alpha_s$. For VP-SDE, let $\alpha(t) = \exp\left(-\frac{1}{2}\int_0^t \beta(s)\,ds\right)$ and $\sigma(t) = \sqrt{1 - \alpha(t)^2}$. For VE/EDM, $\sigma(\cdot)$ denotes the noise scale with $\boldsymbol{x}_t = \boldsymbol{x}_0 + \sigma(t)\,\boldsymbol{\epsilon}$, $\boldsymbol{\epsilon} \sim \mathcal{N}(\boldsymbol{0}, I)$. All random vectors are in $\mathbb{R}^d$, and $I$ is the identity matrix.

## B.1 DDPM (DISCRETE TIME)

**Forward one-step transition.** The DDPM forward transition and the marginal w.r.t. $\boldsymbol{x}_0$ are (Luo (2022, Eq.(31); Eq.(69)-(70))):

$$q(\boldsymbol{x}_t \mid \boldsymbol{x}_{t-1}) = \mathcal{N}\left(\sqrt{\alpha_t}\,\boldsymbol{x}_{t-1},\ (1-\alpha_t)I\right), \quad \text{(DDPM-Forward)} \tag{34}$$

$$q(\boldsymbol{x}_t \mid \boldsymbol{x}_0) = \mathcal{N}\left(\sqrt{\bar{\alpha}_t}\,\boldsymbol{x}_0,\ (1-\bar{\alpha}_t)I\right), \quad \text{(DDPM-Marginal)} \tag{35}$$

with the equivalent reparameterization

$$\boldsymbol{x}_t = \sqrt{\bar{\alpha}_t}\,\boldsymbol{x}_0 + \sqrt{1-\bar{\alpha}_t}\,\boldsymbol{\epsilon}, \quad \boldsymbol{\epsilon} \sim \mathcal{N}(\boldsymbol{0}, I). \quad \text{(DDPM-Reparam)} \tag{36}$$

**Posterior toward $t-1$ (mean and variance).** By linear-Gaussian conditioning,

$$q(\boldsymbol{x}_{t-1} \mid \boldsymbol{x}_t, \boldsymbol{x}_0) = \mathcal{N}\left(\boldsymbol{\mu}_{t-1}^{\text{true}}(\boldsymbol{x}_t, \boldsymbol{x}_0),\ \tilde{\beta}_t I\right), \quad \text{(DDPM-Posterior)} \tag{37}$$

$$\text{where } \boldsymbol{\mu}_{t-1}^{\text{true}}(\boldsymbol{x}_t, \boldsymbol{x}_0) = \frac{\sqrt{\bar{\alpha}_{t-1}}\,(1-\alpha_t)}{1-\bar{\alpha}_t}\,\boldsymbol{x}_0 + \frac{\sqrt{\alpha_t}\,(1-\bar{\alpha}_{t-1})}{1-\bar{\alpha}_t}\,\boldsymbol{x}_t, \quad \text{(DDPM-PosteriorMean)} \tag{38}$$

$$\tilde{\beta}_t = \frac{(1-\alpha_t)(1-\bar{\alpha}_{t-1})}{1-\bar{\alpha}_t}. \quad \text{(DDPM-PosteriorVar)} \tag{39}$$

Equations (38)-(39) from Luo (2022, Eq.(84)-(85), p.12).

**SNR identity.** We will also refer to the signal-to-noise ratio

$$\text{SNR}(t) = \frac{\bar{\alpha}_t}{1-\bar{\alpha}_t}, \quad \text{(DDPM-SNR)} \tag{40}$$

as used to simplify weighting expressions (Luo (2022, Eq.(109), p.14)).

**Two-time mean in unified form.** Taking $\tau = t-1$, the DDPM two-time mean $\boldsymbol{\mu}_{t-1}^{\text{true}}(\boldsymbol{x}_t, \boldsymbol{x}_0)$ equals Equation (38), i.e., the unified form equation (33) with

$$A_{t,\tau} = \frac{\sqrt{\bar{\alpha}_{t-1}}\,(1-\alpha_t)}{1-\bar{\alpha}_t}, \qquad \kappa_{t,\tau} = \frac{\sqrt{\alpha_t}\,(1-\bar{\alpha}_{t-1})}{1-\bar{\alpha}_t}.$$

## B.2 DDIM: TWO-TIME MEAN AND PARAMETERS (SHORT)

Use the same notation as DDPM, let $\alpha_t = 1 - \beta_t$ and $\bar{\alpha}_t = \prod_{s=1}^{t} \alpha_s$. DDIM specifies a non-Markovian reverse conditional whose mean depends on $(\boldsymbol{x}_t, \boldsymbol{x}_0)$ Song et al. (2021a, Eq. 7)):

$$q_\sigma(\boldsymbol{x}_{t-1} \mid \boldsymbol{x}_t, \boldsymbol{x}_0) = \mathcal{N}\left(\underbrace{\sqrt{\bar{\alpha}_{t-1}}\,\boldsymbol{x}_0 + \sqrt{\frac{1-\bar{\alpha}_{t-1}-\sigma_t^2}{1-\bar{\alpha}_t}}\,(\boldsymbol{x}_t - \sqrt{\bar{\alpha}_t}\,\boldsymbol{x}_0)}_{\text{mean}},\ \sigma_t^2 I\right). \tag{41}$$

Hence in our unified linear form, we have

$$\kappa_{t,t-1} = \sqrt{\frac{1-\bar{\alpha}_{t-1}-\sigma_t^2}{1-\bar{\alpha}_t}}, \qquad A_{t,t-1} = \sqrt{\bar{\alpha}_{t-1}} - \kappa_{t,t-1}\sqrt{\bar{\alpha}_t}. \tag{42}$$

**Stochasticity and the $\eta$-parameterization (DDIM Eq. (16)).** A convenient schedule for the reverse variance is

$$\sigma_t(\eta) = \eta\,\sqrt{\frac{1-\bar{\alpha}_{t-1}}{1-\bar{\alpha}_t}}\,\sqrt{1-\frac{\bar{\alpha}_t}{\bar{\alpha}_{t-1}}}, \qquad \eta \in [0,1]. \tag{43}$$

Special cases: (i) $\eta=0 \Rightarrow \sigma_t=0$ gives the deterministic DDIM, where $\kappa_{t,t-1} = \sqrt{\frac{1-\bar{\alpha}_{t-1}}{1-\bar{\alpha}_t}}$ and $A_{t,t-1} = \sqrt{\bar{\alpha}_{t-1}} - \sqrt{\frac{1-\bar{\alpha}_{t-1}}{1-\bar{\alpha}_t}}\sqrt{\bar{\alpha}_t}$; (ii) $\eta=1$ recovers the DDPM variance choice at step $t$.

## B.3  VP–SDE (VARIANCE PRESERVING)

**Forward dynamics and one-time marginals.**  The VP SDE is

$$\mathrm{d}\boldsymbol{x}_t \;=\; -\tfrac{1}{2}\,\beta(t)\,\boldsymbol{x}_t\,\mathrm{d}t \;+\; \sqrt{\beta(t)}\,\mathrm{d}\boldsymbol{w}_t, \qquad t \in [0,1], \tag{VP-SDE}$$

as stated in Eq. (11) of Song et al. (2021b). Let

$$\alpha(t) \;:=\; \exp\!\Big(-\tfrac{1}{2}\int_0^t \beta(s)\,ds\Big), \qquad \sigma(t) \;:=\; \sqrt{1 - \alpha(t)^2}. \tag{44}$$

Solving Equation (VP-SDE) gives the Gaussian marginal

$$p_{0t}(\boldsymbol{x}_t \mid \boldsymbol{x}_0) \;=\; \mathcal{N}\big(\alpha(t)\,\boldsymbol{x}_0,\; (1 - \alpha(t)^2)I\big), \tag{VP-kernel}$$

which appears as the VP case of Eq. (29) in Song et al. (2021b).

**Two-time conditional mean.**  Fix $0 \le \tau < t \le 1$. From the linear solution of equation (VP-SDE), we can write

$$\boldsymbol{x}_t \;=\; \alpha(t)\,\boldsymbol{x}_0 \;+\; \alpha(t)\int_0^t \frac{\sqrt{\beta(s)}}{\alpha(s)}\,\mathrm{d}\boldsymbol{w}_s, \qquad \boldsymbol{x}_\tau \;=\; \alpha(\tau)\,\boldsymbol{x}_0 \;+\; \alpha(\tau)\int_0^\tau \frac{\sqrt{\beta(s)}}{\alpha(s)}\,\mathrm{d}\boldsymbol{w}_s.$$

Conditioned on $\boldsymbol{x}_0$, $(\boldsymbol{x}_\tau, \boldsymbol{x}_t)$ is jointly Gaussian with

$$\mathbb{E}[\boldsymbol{x}_\tau \mid \boldsymbol{x}_0] = \alpha(\tau)\boldsymbol{x}_0, \quad \mathbb{E}[\boldsymbol{x}_t \mid \boldsymbol{x}_0] = \alpha(t)\boldsymbol{x}_0,$$

$$\mathrm{Cov}(\boldsymbol{x}_\tau \mid \boldsymbol{x}_0) = (1 - \alpha(\tau)^2)I, \quad \mathrm{Cov}(\boldsymbol{x}_t \mid \boldsymbol{x}_0) = (1 - \alpha(t)^2)I,$$

$$\mathrm{Cov}(\boldsymbol{x}_\tau, \boldsymbol{x}_t \mid \boldsymbol{x}_0) = \frac{\alpha(t)}{\alpha(\tau)}\big(1 - \alpha(\tau)^2\big)I,$$

where cross-covariance follows Itô isometry: $\mathrm{Cov}(\boldsymbol{x}_\tau, \boldsymbol{x}_t \mid \boldsymbol{x}_0) = \alpha(t)\alpha(\tau)\int_0^\tau \alpha(s)^{-2}\beta(s)\,ds = \frac{\alpha(t)}{\alpha(\tau)}\big(1 - \alpha(\tau)^2\big)I$. Applying the Gaussian conditioning formula—for jointly Gaussian vectors $(y_1, y_2)$, the conditional mean is $\mathbb{E}[y_1 \mid y_2] = \boldsymbol{\mu}_1 + \Sigma_{12}\Sigma_{22}^{-1}(y_2 - \boldsymbol{\mu}_2)$—then yields

$$\boldsymbol{\mu}_\tau^{\mathrm{true}}(\boldsymbol{x}_t, \boldsymbol{x}_0) = \alpha(\tau)\,\boldsymbol{x}_0 + \underbrace{\left[\frac{\alpha(t)}{\alpha(\tau)} \cdot \frac{1 - \alpha(\tau)^2}{1 - \alpha(t)^2}\right]}_{\kappa_{t,\tau}}(\boldsymbol{x}_t - \alpha(t)\,\boldsymbol{x}_0). \tag{VP-TwoTime}$$

Equivalently, in the unified linear form $\boldsymbol{\mu}_\tau^{\mathrm{true}}(\boldsymbol{x}_t, \boldsymbol{x}_0) = A_{t,\tau}x_0 + \kappa_{t,\tau}x_t$,

$$A_{t,\tau} = \alpha(\tau) - \kappa_{t,\tau}\,\alpha(t), \qquad \kappa_{t,\tau} = \frac{\alpha(t)}{\alpha(\tau)} \cdot \frac{1 - \alpha(\tau)^2}{1 - \alpha(t)^2}. \tag{45}$$

*Remark.* With the commonly used linear schedule $\beta(t) = \bar{\beta}_{\min} + t(\bar{\beta}_{\max} - \bar{\beta}_{\min})$, the corresponding $p_{0t}(\boldsymbol{x}_t \mid \boldsymbol{x}_0)$ is given explicitly in Eqs. (32)–(33) of Song et al. (2021b).

## B.4  VE-SDE AND EDM (VARIANCE EXPLODING & $\sigma$-PARAMETERIZATION)

**Forward dynamics and one-time marginals.**  The VE SDE reads

$$\mathrm{d}\boldsymbol{x}_t \;=\; \sqrt{\tfrac{d}{dt}\,\sigma(t)^2}\,\mathrm{d}\boldsymbol{w}_t, \qquad \sigma(0) = 0, \tag{VE-SDE}$$

see Eq. (9) in Song et al. (2021b). It induces the additive-noise marginal

$$p_{0t}(\boldsymbol{x}_t \mid \boldsymbol{x}_0) \;=\; \mathcal{N}\big(\boldsymbol{x}_0,\; \sigma(t)^2 I\big), \tag{VE-kernel}$$

the VE case of Eq. (29) in Song et al. (2021b). $\boldsymbol{x}_t = \boldsymbol{x}_\tau + \sqrt{\sigma(t)^2 - \sigma(\tau)^2}\,\boldsymbol{z}$ with $\boldsymbol{z} \sim \mathcal{N}(\boldsymbol{0}, I)$ independent of $\boldsymbol{x}_\tau$.

**Two-time conditional mean for VE.** With the above coupling, conditioned on $\boldsymbol{x}_0$,

$$\mathrm{Cov}(\boldsymbol{x}_t \mid \boldsymbol{x}_0) = \sigma(t)^2 I, \qquad \mathrm{Cov}(\boldsymbol{x}_\tau \mid \boldsymbol{x}_0) = \sigma(\tau)^2 I, \qquad \mathrm{Cov}(\boldsymbol{x}_\tau, \boldsymbol{x}_t \mid \boldsymbol{x}_0) = \sigma(\tau)^2 I.$$

Thus for $\tau > 0$ and $t > \tau$,

$$\boldsymbol{\mu}_s^{\mathrm{true}}(\boldsymbol{x}_t, \boldsymbol{x}_0) \;=\; \boldsymbol{x}_0 \;+\; \underbrace{\frac{\sigma(\tau)^2}{\sigma(t)^2}}_{\kappa_{t,\tau}}(\boldsymbol{x}_t - \boldsymbol{x}_0) \;=\; \Big(1 - \frac{\sigma(\tau)^2}{\sigma(t)^2}\Big)\boldsymbol{x}_0 \;+\; \frac{\sigma(\tau)^2}{\sigma(t)^2}\,\boldsymbol{x}_t. \quad \text{(VE-TwoTime)}$$

Hence the unified coefficients are $A_{t,\tau} = 1 - \frac{\sigma(\tau)^2}{\sigma(t)^2}$ and $\kappa_{t,\tau} = \frac{\sigma(\tau)^2}{\sigma(t)^2}$.

**EDM uses the same corruption and thus the same two-time mean.** EDM (Karras et al., 2022) parameterizes "time" directly by the noise scale $\sigma$ and corrupts clean data additively: $\boldsymbol{x} = \boldsymbol{x}_0 + \boldsymbol{n}$, $\boldsymbol{n} \sim \mathcal{N}(\boldsymbol{0}, \sigma^2 I)$. Under the same Markov-increments coupling as in VE, the joint $(\boldsymbol{x}_\tau, \boldsymbol{x}_t) \mid \boldsymbol{x}_0$ is Gaussian with the same covariances as above, so the two-time mean of EDM coincides with VE's formula.

