# OpenReview forum: "Temporal  Difference Learning for Diffusion Models"
_ICLR.cc/2026/Conference — ICLR 2026 Conference Desk Rejected Submission_

### Official Review · Reviewer_Zhiz · 2025-10-19

**Soundness:** 2
**Presentation:** 3
**Contribution:** 2
**Rating:** 4
**Confidence:** 4

**Summary:**

This paper proposes a *Temporal Difference (TD)* learning objective for diffusion models. The core idea is that standard diffusion training optimized at isolated time steps does not enforce cross-time consistency, which harms performance for few-step samplers.
The authors recast the denoising trajectory as a *Markov Reward Process (MRP)* and treat training as a policy-evaluation problem, deriving a unified TD objective applicable to both discrete-time and continuous-time diffusion families.
To stabilize optimization across heterogeneous time pairs, they introduce a principled sample-based reweighting scheme $(w_{TD})$.
The TD objective is used as a drop-in regularizer combined with the standard EDM reconstruction loss.
Empirical results on CIFAR-10 (32$\times$32) show consistent FID improvements in few-step (12 -18 step) regimes; gains in standard settings (e.g., large UNet, 18 steps) are present but marginal.

**Strengths:**

1. The paper offers an interesting connection between reinforcement learning (TD / policy evaluation) and diffusion-model training, applying RL tools to enforce  internal  temporal consistency rather than for external reward optimization.

 2. A two-time posterior-mean formulation is proposed, applicable across the DDPM, DDIM, VP, VE, and EDM families, and used to introduce the TD loss.

3. The sample-pair reweighting $w_{TD}$ is introduced to address the differing scales introduced by pairing arbitrary time indices; ablations show its effect on stability and performance.

**Weaknesses:**

1. All experiments are conducted on unconditional CIFAR-10 (32$\times$32), and there is currently no evidence to suggest that the method scales to high-resolution images, conditional generation (e.g., class-conditional or text-conditioned tasks), or other modalities.

2. The paper only compares to baseline EDM and lacks comparison with other training-time methods (Consistency Models, Progressive Distillation), both conceptually and experimentally.  Additionally, the low-NFE gains are demonstrated only with the Heun sampler.

3.  The paper shows mixed results: (1) For the large model at 18 steps, the improvement is from 2.3893 to 2.3768; (2) For the small model at 12 steps, the improvement is from 10.58 to 10.22.  Although these improvements are the paper's main claim, the computational overhead (training time, memory) is not quantified.

**Questions:**

1. Equation (16) supports arbitrary $k$, but why was fix k=1 in all experiments, while the ablation study  for $k \in \{2,3,5\}$ is missing?   Is $k=1$ optimal in practice?


2. Does the proposed method have unique practical advantages and contributions in optimizing information flow, compared to the method in [1]?


3. What is the relationship and specific difference between the proposed *Temporal Difference* and the *Temporal Dynamics*    discussed in [2] and [3]?

[1] Li, S., et al., EVODiff: Entropy-aware Variance Optimized Diffusion Inference, NeurIPS 2025.

[2]. Floros, A., et al., Tracing the Roots: Leveraging Temporal Dynamics in Diffusion Trajectories for Origin Attribution, 2024.

[3]. Guo, X., et al., Dynamical Diffusion: Learning Temporal Dynamics with Diffusion Models, ICLR 2025.

---

> ### Author Response · Authors · 2025-11-25
> **Author Rebuttal**
>
> We thank the reviewer for the constructive and detailed feedback.
>
> Questions regarding computational and memory overhead are addressed in *Common Reply – Computational and Memory Cost section*, and additional experiments are summarized in *Common Reply – Additional experiment section*. A concise discussion of how our TD objective relates to Consistency Models and other training-time methods is provided in *Common Reply – Relation to Consistency Model (CM)*. Below, we address issues that are specific to this review.
>
> ---
> (1) Experimental scope
>
> To address the reviewer’s concern about scope, we have added CIFAR-10 class-conditional experiments and fewer sampling steps evaluations, where TD consistently improves over baseline, and we have run unconditional FFHQ-64×64 experiments under the EDM [4] setup (100M images), again observing consistent FID improvements across 9–18 sampling steps. These results, together with ongoing AFHQv2-64×64 and planned ImageNet-64×64 runs, are summarised in *Common Reply – Additional experiment* and will be incorporated into the revised manuscript.
>
> ---
> (2) Other training-time methods (Consistency Models, Progressive Distillation) and samplers
>
> Our contribution is a **TD-based training objective for base diffusion models**, used from scratch on a single model. Conceptually, this objective is different from Consistency Models and the training-time method is orthogonal to the distillation techniquesuch as Progressive Distillation, hence largely complementary. A concise comparison is given in *Common Reply – Relation to Consistency Model (CM)*, and we will add an explicit discussion of CM and Progressive Distillation (with citations) in the revised related-work section.
>
> For the sampler, we keep it fixed to EDM’s PF-ODE with Heun for focus in our submission, but the formulation is sampler-agnostic.
>
> ---
> 3) Mixed improvements and missing cost quantification
>
> Our method has improvement in 18-step experiment in both conditional and unconditional case of Cifar10, and the gains are significant in compute-constrained regimes, i.e., low NFE, where TD consistently improves FID over the baseline in low NFE setting; detailed numbers, including the additional conditional experiments with fewer sampling steps, are listed in *Common Reply – Additional experiment*. We fully agree that these gains must be considered together with their cost: training with TD introduces a constant-factor overhead, while inference cost is unchanged. The full breakdown is provided in *Common Reply – Computational and Memory Cost* and will be reported in revised manuscript so that the cost-benefit trade-off is clear.
>
> ---
> Answers to specific questions
>
> (1)Why fix k=1? Is this optimal?
>
> Note that in our work, the notion of a k-step return in continuous time is quite different from that in the discrete-time setting. In the latter, k-step has a standard meaning: when computing the TD target, k-step rewards are used and then a k-step bootstrap target is computed. However, in continuous time there is no exact notion of “number of time steps”. The conceptual analogue of the k-step return in the discrete case is actually controlled by the ratio $k/delta$ (intuitively, the larger this ratio, the longer the effective step-return). Since we already sweep over delta, we therefore believe that sweeping over k is not a critical experiment.
>
> ---
> (2) Relation to EVODiff [1] and temporal-dynamics works [2,3]
>
> We view all three cited works as complementary to our TD framework, as they act on different parts of the diffusion pipeline, and we will add an explicit discussion and citations to the related-work section.
>
> EVODiff [1] is an inference-time method: it assumes a fixed pre-trained model and designs entropy-aware ODE solvers to reduce conditional variance during sampling, without modifying the training objective or forward process. In contrast, our method is a training-time objective: we keep the sampler and diffusion family fixed and use a TD loss that enforces multi-step consistency of posterior means along the diffusion time axis, improving fixed-step FID of the base model itself.
>
> “Dynamical Diffusion” [3] targets data-time temporal prediction (e.g., videos, time series) by modifying how current latents depend on previous frames, but still uses standard single-time denoising losses. Our work instead treats the diffusion-time dimension as a trajectory in a Markov reward process and performs TD-style policy evaluation across diffusion steps, without introducing additional temporal dimensions in the data.
>
> Finally, “Tracing the Roots” [2] uses multi-step diffusion trajectories only as features for downstream origin-attribution classifiers, leaving the base model and its training loss unchanged. By contrast, our TD loss is integrated directly into the base model’s training objective to change the learned denoiser itself, with the goal of improving generative quality at fixed NFE rather than solving a forensic classification task.

---

### Official Review · Reviewer_utRC · 2025-10-25

**Soundness:** 2
**Presentation:** 3
**Contribution:** 1
**Rating:** 2
**Confidence:** 4

**Summary:**

This paper introduces a TD learning framework for diffusion model. Instead of the existing method that matches single time-step function evaluation against the ground truth data, the paper proposes to enforce the consistency between two time steps. Further, they propose a reweighting schedule for multiple time-step input. The authors compare the result of TD-assisted training against standard EDM, showing the gain in small NFEs.

**Strengths:**

1. It is not hard to understand the idea. The authors clearly state the contribution of the paper.

**Weaknesses:**

1. The biggest concern of this paper is the significance of the approach: consistency model [1] came out in 2024, which distills the EDM with 1 or 2 time steps. While the proposed method claims the gain in some moderate NFEs, it is still larger than the recent generative model's performance. For my personal view, it is hard to claim the significance of the approach compared to the recently proposed method. Furthermore, the TD framework also reminds me of the consistency distillation, which has been showcased in previous work [1].

2. The gain of the proposed method is also marginal. The proposed method fails to greatly improve EDM.

3. The limited experiment on the CIFAR-10 is also a concern. Nowadays, the generation benchmark has shifted towards more complex distributions like ImageNet. I wonder if the gain of the TD framework is still relevant on that benchmark. The same applies to the architecture, whether the proposed method exihibits gains on the diffusion transformer architecture, too.

4. I wonder the time efficiency of the proposed method, since the evaluation of the loss function requires multiple function calls.



***References***

[1] Consistency Models, ICML 2024.

**Questions:**

See Weaknesses

---

> ### Author Response · Authors · 2025-11-25
> **Author Rebuttal**
>
> We thank the reviewer for the thoughtful comments.
>
> Questions regarding computational and memory overhead are addressed in the *Common Reply – Computational and Memory Cost section*. Additional experiments (CIFAR-10 class-conditional, fewer sampling steps evaluations, and 64×64 extensions) are summarized in the *Common Reply – Additional experiment section*. A concise conceptual comparison with Consistency Models (CM) is given in the *Common Reply – Relation to Consistency Model (CM)*. Below, we respond to points that are specific to this review.
>
> ---
>
> (1) Relation to Consistency Models (CM) [1] and significance
>
> Conceptually, our TD objective is **different and complementary** to Consistency Models (CM). It operates at **training time** on a single base model, relates potentially non-adjacent time indices $(t, t-k)$, and matches **drifts of posterior means** rather than the predictions themselves. A concise summary of these differences is given in the *Common Reply – Relation to Consistency Model (CM)*.
>
> Importantly, our method is a **training-time objective for the base model, not a distillation procedure**: we train a single model from scratch under a TD-based loss (plus a small reconstruction term) and do not perform any teacher–student or few-step student distillation. CM distillation, in contrast, typically assumes a pre-trained teacher and explicitly optimises a student for ultra-low NFE. We will include a clearer discussion of CM and distillation methods in the revised version of the paper.
>
> ---
>
> (2) Experiment result
>
> Our TD-based objective yields consistent gains in the regimes we primarily target. For CIFAR-10 class-conditional with the 55M-parameter ddpmpp UNet, under the same 200M-image budget and sampler, the TD-based objective improves the mean FID over the last $15$% of checkpoints from $2.17$ → $2.12$. Moreover, when we take models trained with an $18$-step schedule on CIFAR-10 conditional and evaluate them with fewer sampling steps ($15$, $12$, $9$), the TD-based model consistently outperforms the baseline, and the relative advantage becomes larger as the number of steps decreases (see *Common Reply – Additional experiment* for detailed numbers).
>
> Beyond CIFAR-10, we have also run unconditional FFHQ-64×64 experiments under the EDM setup (100M training images), where TD again improves FID over EDM across $9–18$ sampling steps (e.g., $3.70$ → $3.41$ at $18$ steps; see Common Reply – Additional experiment). This shows that the TD-based objective remains beneficial at higher resolutions and under tighter step budgets.
>
> In terms of significance, our goal is to show that a **TD-based training objective for base diffusion models** is both conceptually novel and practically useful: by enforcing temporal-difference consistency of posterior means along the diffusion time axis during training, we obtain robust, repeatable improvements over a strong EDM baseline.
>
>
> ---
>
> (3) Scope of experiments beyond CIFAR-10
>
> To address the concern about scope, in addition to the new CIFAR-10 class-conditional results, we have run unconditional FFHQ-64×64 experiments (with EDM settings) and are currently running AFHQv2-64×64 and planned ImageNet-64×64 runs. These extensions, together with our conditional and fewer sampling steps results, are summarized in the *Common Reply – Additional experiment section*. We believe our experiments so far are well-designed and sufficient to support our main claims in this submission.
>
> ---
>
>  [1]Song, Yang, et al. "Consistency models." (2023).

---

### Official Review · Reviewer_8bya · 2025-10-30

**Soundness:** 3
**Presentation:** 3
**Contribution:** 3
**Rating:** 6
**Confidence:** 2

**Summary:**

This paper introduces a temporal difference (TD) learning objective to address the lack of cross-time consistency in standard diffusion model training, which typically relies on single-step reconstruction losses. The authors derive a unified TD loss that enforces multi-step consistency along the trajectory and can be applied to both discrete and continuous-time diffusion formulations. Empirical results on CIFAR-10 demonstrate that combining this TD objective with standard training improves sample quality, particularly under low-compute sampling conditions.

**Strengths:**

- This paper is original. While recent works have framed diffusion sampling as an RL problem (e.g., DDPO, DPOK) to optimize black-box rewards, this paper takes a distinct path that it reformulates the diffusion training itself as a ​policy evaluation​ problem within a Markov Reward Process (MRP). The key insight is to define a "reward" as the drift of the true posterior mean and a "value function" as the cumulative drift, leading to a Temporal Difference (TD) loss that directly penalizes inconsistencies in the model's denoising trajectory.
- As far as I checked, this paper is technically sound. The authors carefully derive it starting with a discrete-time MRP formulation and then elegantly extending it to continuous time. A particularly strong aspect is the derivation of a ​principled, sample-based reweighting scheme​ (w_TD) to stabilize training.
- The empirical evaluation, while limited to CIFAR-10, is rigorous. It includes ablations on key hyperparameters (mixing coefficient, stride, weighting), and the results consistently support the claims. The fact that the method provides a clear advantage in the low-NFE regime strengthens the validity of the approach.
- The paper is generally well-written and clear, especially given the technical complexity of bridging these two domains. The use of a ​unified notation​ (Table 1) for different diffusion families is a major aid to understanding and helps demystify the differences between model types. Algorithm 1 provides a concrete summary of the training procedure.

**Weaknesses:**

- This paper's central claim is that the TD approach is a "general drop-in" for a "wide range of diffusion generative models." However, the experimental validation is confined exclusively to ​unconditional image generation on CIFAR-10 (32x32)​​ using a single architecture (SongUNet) and a single sampler family (EDM's probability-flow ODE).
  - CIFAR-10 is a small, well-controlled dataset that is not representative of the complex, high-resolution data where diffusion models are most prominently used and where temporal consistency issues might be more pronounced. The failure to demonstrate the method on even a standard benchmark like ImageNet 64x64 or a conditional task (e.g., text-to-image) significantly weakens the claim of generality.
- This paper mentions the "additional constant-factor computational and memory overhead" as a limitation but provides no data to quantify this cost. For a method aiming to improve efficiency, this overhead is a critical practical factor.
  - We cannot evaluate the practical utility of the method without understanding its cost. How much does the target network and the additional forward/backward passes impact compute/memory usage? This lack of information makes it impossible to conduct a fair cost-benefit analysis.
- The MRP formulation, while elegant, is presented more as an analogy than a deep theoretical connection. The "reward" is defined as a vector difference of posterior means, and the "value function" is the cumulative drift. This differs significantly from standard RL, where rewards are scalar and value functions represent expected cumulative reward.

**Questions:**

How does your TD approach conceptually and empirically relate to Consistency Models (CM) [Song et al., 2023]? Both methods aim to improve temporal consistency, but CMs enforce a strict consistency function. Could you discuss the trade-offs? For instance, does your method offer advantages in sample quality or training stability compared to CM distillation, especially in the very low NFE regime (e.g., NFE < 10)?

Could you provide concrete numbers for the training time and memory usage increase (%) when adding the TD objective compared to the baseline EDM, for both the small and large model configurations?

---

> ### Author Response · Authors · 2025-11-25
> **Author Rebuttal**
>
> We thank the reviewer for the careful reading and helpful comments.
>
> Questions regarding computational and memory overhead are addressed in the *Common Reply – Computational and Memory Cost section*. Additional experimental results beyond unconditional CIFAR-10 (including CIFAR-10 class-conditional, fewer sampling step evaluations, and 64×64 experiments) are summarized in the *Common Reply – Additional experiment section*. A concise comparison between our TD objective and Consistency Models (CM) is provided in the *Common Reply – Relation to Consistency Model (CM)*. Below, we address points that are specific to this review.
>
> ---
>
> (1) “general drop-in” and Experiment
>
> Our intention with the “general drop-in” is to emphasise **formulation-level generality**: the TD loss is defined purely in terms of the two-time posterior means of the diffusion family, and therefore applies on DDPM/EDM-style models at different resolutions and with different conditioning mechanisms (class-conditional, text-conditioned, etc.), without specifying architecture or sampler.
>
> To address your concern about empirical evidence, we have added CIFAR-10 class-conditional experiments and run unconditional FFHQ 64×64 experiments (with EDM settings), and we are also running AFHQv2-64×64 and planned ImageNet-64×64 runs; the setups and results are summarised in the *Common Reply – Additional experiment section* and will be incorporated into the revised manuscript.
>
> ---
>
> (2) MRP / RL connection and vector-valued rewards
>
> This is precisely one of the novelties of our reformulation of the diffusion process. To better connect our formulation with conventional RL, we can view it as evaluating a single policy under many different reward functions—one per pixel—yielding the same number of value functions. Note that this also differs from prior work we reviewed that casts diffusion-model training as a policy-gradient problem with hand-crafted rewards tailored to specific goals. In contrast, our approach enforces consistency across time steps without any reward customisation, because the reward is fixed to be always well-defined across settings.
>
> ---
>
> (3) Relation to Consistency Models [2] (CM) and low NFE
>
> Conceptually, our TD objective differs in both **which time indices** it relates and **what** it matches (drifts of posterior means instead of predictions themselves), and is therefore largely **complementary** to CM. A detailed comparison is given in the *Common Reply – Relation to Consistency Model (CM)*.
>
> Importantly, our method is a **training-time objective for the base model**, not a distillation procedure: we train a single model from scratch under a TD-based loss (plus a small reconstruction term) and do not perform any teacher–student distillation or few-step student training. Thus, it is not a like-for-like comparison to directly benchmark our method against CM distillation. Instead, we view CM-style distillation as a potentially complementary stage that could be applied on top of a model trained with TD.
>
> We also report evaluations for fewer sampling steps, where the relative improvement of TD over the EDM baseline becomes larger as the number of steps decreases; detailed results are provided in the *Common Reply – Additional experiment section*.
>
> ---
>
>
> [1] Karras, Tero, et al. "Elucidating the design space of diffusion-based generative models." Advances in neural information processing systems 35 (2022): 26565-26577.
>
> [2]Song, Yang, et al. "Consistency models." (2023).

---

### Official Review · Reviewer_hRXk · 2025-11-03

**Soundness:** 2
**Presentation:** 2
**Contribution:** 2
**Rating:** 4
**Confidence:** 4

**Summary:**

This paper proposes a TD-learning based method for training diffusion models. Numerical experiments are conducted to demonstrate the effectiveness of the proposed method. Ablation studies are also included.

**Strengths:**

1. This paper introduces TD-learning method for training the score network.
2. The authors conducted numerical experiments for their methods.

**Weaknesses:**

I have several major concerns:
1. The problem studied in this paper is not clearly stated. The authors cited RL-based approaches for post-training/fine-tuning of diffusion models while they compare their approach with EDM in numerical study. This makes me confused the goal of this work.
2. Following 1), if the goal is fine-tuning, the authors should include comparison to other methods such as DPOK. If the goal is pre-training of score networks, I am very concerned the computational cost of adding a RL block in this stage. At least from experiments, the performance gap compared to EDM is not that significant.
3. The interpretation of (14) is not that convincing to me. One can rewrite (14) as $ \delta_t  = (\mu_{t-1}^{true} - \mu_{\theta, t-1}) - (\mu_{t-2}^{true} - \mu_{\theta', t-2}) $. In this way, forcing $ \delta_t $ small is forcing all the score matching error small *uniformly* over all time steps. I am not sure if this is possible theoretically and if this is indeed necessary. I suggest the authors carefully elaborate this point.

**Questions:**

One minor question:
1. Why do you still add EDM loss to the final loss function? Could you explain how much could you gain with and without this term?

---

> ### Author Response · Authors · 2025-11-25
> **Author rebuttal**
>
> We thank the reviewer for the detailed and constructive feedback.
>
> Questions regarding computational and memory overhead are addressed in the *Common Reply – Computational and Memory Cost section*. Below, we address points that are specific to this review.
>
> ---
>
> (1) Clarifying the goal and positioning of our paper
>
> Our goal is **not** post-training or alignment fine-tuning (as in DDPO [1], DPOK [2], etc.), but to introduce a **new formulation and objective for base diffusion model training** that builds with a temporal-difference (TD) loss. Concretely:
>
> - In our Markov reward process formulation, both the “reward” and “value” are fully determined by the known forward corruption process and do not depend on any external or task-specific reward model. Our proposed algorithm is a value-based policy evaluation algorithm, not a control one.
>
> - We start from a standard diffusion model setup and **train the model from scratch**
>
> - The **core loss** we optimize is the TD loss, and the motivation is to enforce multi-step temporal consistency of posterior means along the diffusion time axis.
>
> - We additionally include the diffusion model’s reconstruction loss with a small weight as a regularizer, to stabilise optimisation (see point (2) below). The whole network is trained from scratch under this combined objective.
>
> In contrast, the existing RL-based methods (such as DDPO [1], DPOK [2]) operate in a different regime (e.g., text-conditioned, human-trained scalar rewards, policy-gradient updates for alignment). Since we neither use external rewards nor perform RL fine-tuning, we view such methods as orthogonal rather than direct baselines. Studying the utility of our approach in fine-tuning could be a future effort.
>
> We will revise the introduction to make it explicit that our contribution is a TD-based training objective for base diffusion models.
>
> ---
>
> (2) Why do we add EDM [3] loss to the final loss function?
>
> From both theoretical and practical perspectives, TD with nonlinear function approximation—especially under non-stationary training—does not, in general, come with a guarantee of convergence to a unique fixed point. The non-stationarity here typically arises because the network parameters update, which alters the transition probabilities, and the usage of the target network. (By contrast, with linear function approximation and on-policy sampling, TD(0) converges to a unique projected Bellman fixed point under standard assumptions). In this context, EDM can be viewed as a regularizer that biases learning toward the EDM solution: i.e., the combined objective is constructed so that any minimizer of the regularized objective is also an EDM solution (e.g., under common over-parameterization assumptions of NNs). This interpretation does not, by itself, assert global convergence guarantees for nonlinear TD; it only clarifies the relationship between the regularized objective and the EDM optimum.
>
> ---
>
> (3) Interpretation of TD loss and the “uniform small error” concern
>
> To interpret the relationship between our method and “uniform small error” the reviewer mentioned, we explain under the discrete case. Eq. (14) expresses the TD error as the difference between the one-step diffusion drift and the one-step model drift of posterior means.
>
> One can indeed algebraically rewrite $\delta _ t$​ as the difference of  two “score-matching errors” at times $t-1$ and $t-2$, but minimizing $\|\delta _ t\|^2$ does not require that all score errors be uniformly small across all times. Instead, it enforces that the difference between consecutive errors is small, i.e., that the model’s change from $t$ to $t-1$ matches that of the true diffusion process. In this sense, the TD loss controls relative temporal consistency rather than absolute uniform accuracy over time.
>
> ---
>
> [1] Black, Kevin, et al. "Training diffusion models with reinforcement learning." arXiv preprint arXiv:2305.13301 (2023).
>
> [2] Fan, Ying, et al. "Dpok: Reinforcement learning for fine-tuning text-to-image diffusion models." Advances in Neural Information Processing Systems 36 (2023): 79858-79885.
>
> [3] Karras, Tero, et al. "Elucidating the design space of diffusion-based generative models." Advances in neural information processing systems 35 (2022): 26565-26577.

---

### Author Response · Authors · 2025-11-24
**Author Rebuttal**

**Common Reply**

---

### Computational and Memory Cost

Several reviewers (hRXk, 8bya, utRC, Zhiz) asked about the computational and memory overhead of using our TD Method.
Our method only changes the training objective: the TD loss (plus a small reconstruction term) is applied during training, while the inference-time sampler, NFE, and the memory remain identical to the baseline diffusion model for a fixed sampler and step schedule. Therefore, the computation and memory are constant additions and identical in Big-O notation.

The extra training cost comes from (i) evaluating the (target) network at a second time point and (ii) computing two posterior means per sampled time pair, which yields a constant-factor training-time overhead. For example, on CIFAR-10 class-conditional with the 55M-parameter UNet trained for 200M images, the baseline training takes 50h 33m 30s (45.7 s/tick), while training with TD takes 71h 36m 48s (64.7 s/tick) on the same hardware. CPU memory increases from 2.29 GB to 2.74 GB (≈20%), and GPU memory from 16.58 GB to 17.51 GB (≈6%).

We will report these measurements, together with analogous numbers for the unconditional small/large models, in the revised manuscript so that the cost-benefit trade-off (moderate, training-only overhead vs. consistent FID gains at fixed NFE) is clear.

---

### Relation to Consistency Model(CM)[1]

Multiple reviewers (8bya, utRC, Zhiz) asked how our TD objective related to Consistency Models (CM) [1] and the distillation method. Both the motivation and approach are quite different and largely complementary:
Our method focuses on training-time instead of distillation.

- *Time indices*. CM always relates adjacent noise levels, typically a pair $(t, t-1)$. Our TD formulation can use a general pair $(t, t-k)$ (or the continuous-time analogue), i.e., observations that are $k$ steps apart along the diffusion timeline, enforcing consistency of the implied drift over longer spans.

- *What is matched in the objective.* A typical CM loss directly matches predictions of $x_0$ at two time steps: $\left\| \hat{x} _ 0(x _ t) - \hat{x} _ 0(x_{t-1}) \right\|^2$。 Our TD loss instead matches differences of posterior means (drifts):
$\left\| (\mu^{\text{true}} _ {t-\tau} - \mu^{\text{true}} _ {t-k-\tau}) - (\hat{\mu} _ {t-\tau} - \hat{\mu} _ {t-k-\tau}) \right\|^2$
i.e., CM enforces that the functions $\hat{x}_0(\cdot)$ themselves agree across time, while TD enforces that the change in posterior means between two times matches the true diffusion drift. These objectives are not equivalent and fundamentally different; to a large extent, they are orthogonal and could be combined.

Overall, our TD loss is used directly when training the base model from scratch by matching the diffusion drift and the model drift. For these reasons, we view CM and other paper reviewers mentioned as complementary to our TD-based training rather than direct competitors.

[1] Song, Yang, et al. "Consistency models." (2023).

---

### Author Response · Authors · 2025-11-25
**Author Rebuttal**

**Common Reply (cont.)**

---

### Additional experiment

Several reviewers (8bya, utRC, Zhiz) requested evidence beyond unconditional CIFAR-10. Our TD loss is defined purely via the diffusion family’s two-time posterior means and is agnostic to conditioning and backbone, so it also applies to class-conditional and text-conditional models and to higher resolutions.

**Conditional Models.** To support this claim, we have added CIFAR-10 class-conditional experiments using a 55M-parameter UNet and the same training setting as EDM [2]. Under the same training budget (200M images) and sampler (PF-ODE with Heun), the TD-based objective consistently improves over the baseline; when tracking FID over training and averaging the last $15$% of checkpoints at 18 sampling steps, we obtain $2.170$ for EDM vs $2.120$ for our TD method. These results show that the relative improvements we observe for unconditional CIFAR-10 also hold in the conditional setting.

**Fewer-step Inference.** In the CIFAR-10 class-conditional setting, we also performed an additional robustness test across different NFEs. We took models trained with an $18$-step schedule and evaluated them with **fewer sampling steps** ($15$, $12$, and $9$), and again computed the mean FID over the last $15$% of checkpoints. The TD-based model consistently outperforms the baseline **at all step counts**, and the relative advantage becomes larger as the number of steps decreases:

- 18 steps: baseline $2.170$ → TD  $2.120$
- 15 steps: baseline $2.311$ → TD  $2.235$
- 12 steps: baseline $2.365$ → TD  $2.270$
- 9 steps: baseline $4.115$ → TD  $3.799$

This further supports our claim that TD is particularly beneficial when using fewer sampling steps.

**Higher resolution images.** To test scalability to higher resolutions, we have also run unconditional FFHQ 64×64 experiments using the EDM [4] setup, with 100M training images and the same sampler (PF-ODE with Heun). Again, the TD-based objective consistently improves over EDM across all step counts (mean FID):

- 18 steps: baseline $3.703$ → TD $3.415$
- 15 steps: baseline $4.077$ → TD $3.696$
- 12 steps: baseline $4.820$ → TD $4.604$
- 9 steps:  baseline $8.233$ → TD $7.563 $

These FFHQ results indicate that the improvements from TD extend beyond CIFAR-10 to higher-resolution data, while preserving the advantage with fewer sampling steps.

**Additional running experiments.** Due to computational limits in the rebuttal period, we cannot yet provide full ImageNet-64×64 numbers. As additional 64×64 complements, we are currently running experiments on AFHQv2 [3] (64×64) following the EDM [4] setups, and we are running and plan to include ImageNet-64×64 results in the revised version or camera-ready. However, we do believe that our experiments so far are well-designed and sufficient to support our main claims in this submission.


---

[2] Karras, Tero, Samuli Laine, and Timo Aila. "A style-based generator architecture for generative adversarial networks." Proceedings of the IEEE/CVF conference on computer vision and pattern recognition. 2019.

[3] Choi, Yunjey, et al. "Stargan v2: Diverse image synthesis for multiple domains." Proceedings of the IEEE/CVF conference on computer vision and pattern recognition. 2020.

[4] Karras, Tero, et al. "Elucidating the design space of diffusion-based generative models." Advances in neural information processing systems 35 (2022): 26565-26577.

---

### Note · Program_Chairs · 2026-01-17
**Submission Desk Rejected by Program Chairs**

The following references in this submission do not refer to real documents and/or have major errors in bibliographic information:

 Jialin Chen, Zhiqing Sun, Jing Shi, Yi Ren, and Zhou Zhao. Diffusion model loss-guided reinforcement learning for text-to-speech synthesis. arXiv:2405.14632, 2024.